# Measurement and Simulation of Risk Coupling in Port Hazardous Chemical Logistics

**DOI:** 10.3390/ijerph20054008

**Published:** 2023-02-23

**Authors:** Xiaoxiao Geng, Yongwei Lv, Li Zhao, Yingchen Wang

**Affiliations:** 1Architecture and Art School, Hebei University of Engineering, Handan 056038, China; 2Management Engineering and Business School, Hebei University of Engineering, Handan 056038, China

**Keywords:** system dynamics, port hazardous chemicals logistics, risk coupling effect, simulation study

## Abstract

Hazardous chemical logistics and transportation accidents are the main type of port safety accidents. Correctly and objectively analyzing the causes of port hazardous chemical logistics safety accidents and the coupling mechanisms of risk generation are very important for reducing the occurrence of port hazardous chemical safety accidents. Based on the causal mechanism and coupling principle, in this paper, we construct a risk coupling system for port hazardous chemical logistics and analyze the coupling effects in the risk system. More specifically, a personnel–ship–environment–management system is established and the coupling between the four systems is explored. Taking Tianjin Port as an example, the risk coupling factors are analyzed in combination with system dynamics simulation. Under dynamic changes in coupling coefficients, the change of coupling effects are explored more intuitively, the logical relationships between logistics risks are analyzed and deduced, a comprehensive view of the coupling effects and their evolution process in accidents is provided, and the key causes of accidents and their coupling risk effects are identified. For port hazardous chemicals logistics safety accidents, the presented results not only allow for effective analysis of the causes of safety accidents, but also provide reference for the formulation of prevention strategies.

## 1. Introduction

With the development of the world economy and international trade, the development of port logistics has entered a new stage. Prospect and demand changes in the logistics industry have forced the development of port logistics to conform to the present situation of economic development, giving full play to its leading role and better promoting the rapid development of port economy. As ports are important links in the logistics and transportation chain of dangerous chemicals, it is extremely urgent to manage the logistics and safety of transportation. The storage and transportation of dangerous chemicals in ports involve logistics and transportation work characterized by complex operation processes, strong safety requirements, high technical standards, and a high degree of equipment specialization. The logistics associated with dangerous chemicals have appeared with the development of the chemical industry, with the main service objects being enterprises that produce and circulate dangerous chemicals.

Definition and types of dangerous goods: As far as maritime transport is concerned, according to the international regulations on dangerous chemicals, dangerous goods can be divided into nine categories: explosives, gases, flammable liquids, flammable solids, oxidants, toxic and infectious materials, radioactive materials, corrosive materials, and sundries. Bulk dangerous goods include bulk liquid dangerous goods (such as petroleum, liquefied gas, bulk chemicals, etc.) and bulk dangerous goods (such as mineral powder, seed cake, ammonium nitrate, etc.).

The size of the risk depends on the scale of the loss and the possibility of occurrence. When a risk brings little loss and the probability of this loss is low, we can regard it as a low-risk situation; if the risk leads to huge losses, and there may be losses of people and money, we can regard it as a high-risk situation. The level of risk also depends on the individual’s sensitivity to risk. Some people regard risk as high-risk and others as low-risk, so everyone’s risk level will be affected.

Risk is an uncertain behavior; however, the size of the risk determines the scale and possibility of the loss. Risk is objective and measurable; risk refers to the probability and severity of accidents.

With the acceleration of industrialization in China, the demand for hazardous chemicals is increasing, with the considered hazardous chemicals being complex and dangerous. As an important transportation hub connecting the land, sea, and waterways, ports have a strong carrying capacity for hazardous chemicals. Furthermore, ports are loaded, unloaded, and trans-shipped frequently. Therefore, the logistics and transportation of hazardous chemicals in ports are extremely risky, for which a comprehensive, efficient, and perfect risk evaluation system must be established; furthermore, the risk associated with the port’s hazardous goods logistics must also be analyzed.

Scholars at home and abroad have made some achievements in research focused on hazardous chemical logistics. Xiao et al. (2017), based on an analysis of the safety risks of hazardous chemical transportation at home and abroad, studied the four main factors affecting the safety risks of road transportation. Then, they proposed a hybrid method comprising a dynamic fault tree and Bayesian network to identify and evaluate the dynamic safety risks of hazardous chemical road transportation [1]. Gul et al. (2019) proposed a risk assessment model for the transportation routes of oil transportation enterprises based on fuzzy mathematics [2]. Wu Y et al. (2019) integrated their research progress to build an evaluation system based on the nature and risk tolerance of hazardous chemicals, which allows for study of the risk of hazardous chemicals leaking into the sea [3]. Liu Y X (2019) comprehensively analyzed the storage safety of hazardous chemicals using the fault model and impact analysis method, and found that the key factors affecting storage safety were a lack of fire control facilities, excessive storage in the warehouse, and unsafe operation by operators. Based on this, corresponding countermeasures were put forward, and the authors stated that further research should include quantitative analysis to make the conclusion more accurate and scientific [4]. As far as the current research situation is concerned, most relevant scholars at home and abroad have identified and evaluated the risks associated with the transportation of hazardous chemicals, ignoring the cross-influence of risk systems [5,6].

Related research on risk coupling has been widely conducted in various fields, and various methods have been adopted to analyze the coupling effect. Yang Z Q (2017) established a risk coupling model for hazardous chemicals based on the coupling principle in order to analyze the coordination degree and put forward constructive suggestions [7]. Chen W K (2017) constructed a hierarchical relationship model of risk factors by explaining the structural model and obtained a flow chart of the relationship among the risk factors of explosion accidents of dangerous chemicals through coupling analysis [8]. Wang H (2018) used ANSIs Workbench software 2017 to simulate the rocking effect of a liquid tanker, as well as TruckSim software (2023) to simulate the fluid–solid coupling of the liquid tanker, in order to carry out numerical simulations [9]. Jiang et al. (2018) used trigger theory to analyze the failure scenarios of other tanks in a storage tank area caused by the coupling of multiple disasters and analyzed the coupling effect of multiple disasters [10]. Li Q H (2020) used accident cause theory and coupling theory to build a Bayesian network following the framework of people, vehicles, roads, and environment combined with machine learning. The established model was analyzed and evaluated by Bayesian network reasoning, and important factors affecting the accident risk in the road transportation of hazardous chemicals were obtained and suggestions were put forward [11]. Shi S.L (2021) as studied the coupling relationships of risk factors in the transportation of dangerous chemicals and constructed four main risk factor index systems for the comprehensive evaluation of risk factors (human, material, environment, and management), and organically combined AHP (Analytic Hierarchy Process) with a coupling degree model to construct an evaluation model for the coupling degree of the road transportation of dangerous chemicals [12]. Wang H X (2021) analyzed the coupling between risk factors and risks from three aspects—single, double, and multi-factors—and established a risk coupling measurement model by applying the N-K model, then applied this model to analyze risk coupling [13]. Ma J W (2022) took the trigger principle as reference to analyze the triggering process of water traffic accidents, quantitatively studied the coupling relationship of four risk systems (i.e., people, ships, environment, and management) in the water traffic system, and analyzed and measured the risk coupling factors of water traffic accidents [14]. Zhao C T (2022) identified the risk factors affecting the safe operation of helicopters and, based on this, analyzed the risks using MATLAB. In this way, the risk coupling coefficients of single, double, and multiple factors could be obtained [15]. Based on the risk coupling relationship of four systems—people, vehicles, roads, and the environment—Su (2022) built a traffic accident risk coupling network analysis model using the N-K model, which allows for quantitative calculation of the probability of risk occurrence, obtaining the risk coupling values under various coupling modes, and calculating the risk size under various coupling modes [16]. At present, research on the coupling effect of risks has mainly been applied to the field of safety accidents. Most scholars have used a single method to analyze the coupling effect of risks, neglecting the interactions and coupling effect of many factors in safety accidents [17,18]. Moreover, the research of risk coupling mainly focuses on road transportation of dangerous chemicals, safety accidents, and maritime traffic accidents, ignoring the importance of ports as transit transportation.

Scholars at home and abroad have made great achievements in the research of system dynamics, and the simulation of risk factors based on the principle of system dynamics has become a hot research direction. Chen (2021) constructed supply chain models of building materials in two environments, then established system dynamics flow charts using the Vensim PLE simulation software 5.9c [19]. Ma H (2021) analyzed the interactions and causal relationships between social factors and individual risk perception, classified multiple feedback loops, established a system dynamics model, and simulated the evolution of risk perception in different group environments through a series of experiments [20]. Qiao W G (2019) studied the causes of coal mine accident risk coupling from the perspective of homogeneity and heterogeneity, according to the different participating factors. On this basis, the causal relationships between mine disaster risk causes and event causes were determined, and a non-linear dynamic coupling model was established to measure the risk cause coupling relating to mine disasters [21]. Sun G S (2021), based on system dynamics, simulated the factors affecting the productivity of green building enterprises from the perspectives of the project, management, human resources, and technology, and gave corresponding countermeasures and suggestions [22]. Patricia (2021) used a qualitative research method to select 13 indicators from an index system, used a quantitative method and causality diagram to draw the system flow diagram, gave the dynamic equation of each variable, and verified and simulated the obtained results [23]. Based on system science theory and evolutionary game theory, Zhou (2021) constructed an evolutionary game and system dynamics model for the supervision mechanism of household waste sorting behavior, determined the stable equilibrium point of the system using Vensim software simulation 5.9c, and discussed the sensitivity of system evolution when the parameters changed [24]. System dynamics has been used by scholars to conduct simulation research on the causes of safety accidents, but few studies have focused on simulation research of logistics and transportation safety accidents [25].

Through a literature analysis, it can be found that domestic and foreign scholars have studied the risk problems from different angles, according to different subjects, where the analysis of risk coupling mainly focuses on the measurement of its coupling degree. Through a literature review, we found that few articles have considered the risk coupling of port hazardous chemical logistics; however, in the actual hazardous chemical logistics and transportation at ports, the coupling effect of risk systems such as personnel, ships, environment, and management can bring great harm and far-reaching influence. Therefore, it is necessary to control the risk coupling effect of port hazardous chemicals logistics; in particular, measurement of the multi-factor coupling effect is difficult, and the process of its generation and accumulation is difficult to observe intuitively. The study of risk coupling by system dynamics simulation is still in its infancy, and few scholars have combined the logistics risk of hazardous chemicals in ports with system dynamics simulation. At present, the related research on port hazardous chemical logistics has yet not addressed these problems.

Considering the above, it is of great research value to conduct risk coupling analysis focused on ports, as they are important distribution centers of hazardous chemicals, as well as adopting system dynamics to simulate logistics safety accidents. Therefore, aiming at the multi-factor coupling problem of port hazardous chemical logistics risk, we construct a multi-factor coupling measurement model, which is then combined with a computer simulation method based on system dynamics risk coupling measurements to simulate and analyze the above-mentioned risk coupling model, in order to reveal the complex dynamics of the process of port hazardous chemical logistics risk coupling, as well as to deeply explore the risk coupling of port hazardous chemical logistics, which has certain theoretical and practical significance.

The contributions of this paper are as follows.

(1)We construct an index system for port hazardous chemical logistics risk, point out the coupling relationships among various risk factors, and lay a solid theoretical foundation for further in-depth analysis of the risk coupling mechanisms of port hazardous chemical logistics.(2)The coupling relationships among the four risk systems (i.e., man, ship, environment, and management) are analyzed, and a hierarchical network model and causal flow chart among the risk systems are established. The N-K model is used to measure the coupling effect between two or three factors, and the influence of the risk system on the coupling effect is analyzed, which has certain theoretical and practical value for research on the logistics risk associated with hazardous chemicals in China’s ports.(3)The system dynamics software Vensim PLE 5.9c is used to simulate the risk coupling. By changing the degree of influence between the coupling effects in the variable analysis, we can determine the change in the coupling time of system risk caused by two-factor coupling. Among three factors, we adjust the coupling variables of the personnel–ship–management system through changing a single system variable or several system variables. Finally, combined with the simulation results under different conditions, some suggestions for risk coupling control are given.

The remainder of this paper is organized as follows. In Section 2, the coupling effect of port hazardous chemical logistics risk is introduced. In Section 3, the N-K model is constructed to measure the coupling degree. In Section 4, the coupling effect is studied through system dynamics simulation. Finally, Section 5 provides our conclusions.

## 2. Analysis of the Coupling Effect of Risk Factors in Port Hazardous Chemicals Logistics

### 2.1. Port Hazardous Chemicals Logistics Risk System Analysis

#### 2.1.1. Port Hazardous Chemicals Logistics Safety Accidents

The main sources of accident information are direct accident information, accident investigation reports, port accident books, accident literature published by local port management departments, and typical domestic accidents information searchable on the Internet. When sorting out accident data, our main principle of collecting accidents was to focus on collecting safety accidents in the port area, mainly including accidents related to hazardous chemical terminals, hazardous chemical storage, petrochemical storage and transportation areas, petrochemical terminals, dangerous goods storage yards, roads in the port, and so on. To reflect the basic composition of the collected port accident data, the number of accidents in each year in the accident data were counted, and the results are shown in Figure 1 [26].

As can be seen from Figure 1, the number of safety accidents related to hazardous chemicals in ports have increased over the past ten years; in recent years, the number of safety accidents reached more than 20. The accidents caused by hazardous chemicals are very harmful; they not only can cause the loss of lives and property, but may also bring irreversible harm to the ecological environment. Therefore, it is very important to pay attention to the safety of logistics and transportation of hazardous chemicals in ports, and it is necessary to control relevant risks to reduce the accident rate.

#### 2.1.2. Building a Logistics Risk System for Hazardous Chemicals in Ports

Port hazardous chemicals logistics is a special supply chain system that covers the transportation, storage, distribution, and retail of hazardous chemicals, which requires the safety management of hazardous chemicals. The safety management of dangerous chemicals in ports should be carried out from the beginning of ship transportation into the port, and every link of the supply chain is considered very important, from the loading and unloading of containers for goods entering port to transportation into a reserve warehouse. A flow chart of logistics and transportation in a port hazardous chemicals terminal is provided in Figure 2.

In the process of loading and unloading hazardous chemicals in ports, risk factors inevitably arise. Every shipment link is inseparable from the operations of personnel and transportation by ships. At the same time, transportation is also affected by environmental factors and management departments. Therefore, we divided the whole logistics transportation process into four risk systems: personnel, ships, environment, and management. Risk coupling means that the occurrence and function of one kind of risk depends on and influences other risks when a system is carrying out activities [27]. The risk factors in the four risk systems of logistics transportation are inter-related and interact with each other, thus changing the evolution process of risk factors and forming risk coupling in the process of risk evolution.

Based on the practice and related research of port hazardous chemical logistics risk management, the influencing factors of port hazardous chemical logistics risk can be classified into four categories; namely, the personnel risk system, ship risk system, environmental risk system, and management risk system. These are detailed in the following.

Personnel risk factors

Personnel factors are the main reason for safety accidents relating to dangerous chemicals logistics and transportation in port, comprising production, processing, inspection, maintenance, loading, and unloading personnel. Relevant risk and safety accidents in port hazardous chemical logistics mainly involve illegal operations, the knowledge and quality of operators, professional skills, safety awareness, unstable psychological quality, proficiency in equipment operation, fatigue, sense of responsibility and staff discipline, and so on.

Ship risk factors

There are two main risk factors in the transportation of dangerous chemicals: the dangerous chemicals themselves, and the transportation facilities and equipment. The physical characteristics of hazardous chemicals are easily affected by the external environment, and perfect logistics and transportation equipment can effectively prevent the occurrence of hazardous-substance-related accidents. In this line, the risk factors of ships and aircraft include the characteristics of the hazardous chemicals, the loading state, the functions of important components, equipment maintenance, aging and wear of facilities and equipment, and equipment failure.

Environmental risk factors

Hazardous chemicals are sensitive to the environment, easily leading to fires and explosions under certain conditions. Environmental risk factors include the indoor and outdoor working environment, waterway traffic conditions, traffic density, fire protection grade, safety signs, and so on.

Management risk factors

Accidents may also be caused by management factors. Transportation and logistics, enterprise management, overlapping of supervision scope and power, and emergency rescue command and decision making all have a certain impact on the efficiency of the management department. Management risks mainly include the established safety system, personnel safety training, safety reward mechanism, investment in monitoring and early warning, safety protection devices, preparation of emergency plans and drill management, safety culture and education management, and so on [28].

Enterprise safety management can not only reduce the incidence of accidents, but may also reduce the consequences of accidents. In the stage of accident risk prevention, a group of excellent management talents should be trained. Therefore, the key to controlling safety accidents in port hazardous chemical logistics and transportation is the management factor (Figure 3).

### 2.2. Analysis of Coupling Effect of Port Hazardous Chemicals Logistics Risk

The term “coupling” has been defined in physics as “the interdependence between two or more entities”. In terms of the overall system theory, coupling is a system. Everything in this world is connected, which is what we call a system. Moreover, there exist linear and non-linear interactions between systems, which we can call coupling [29].

The coupling of port hazardous chemicals logistics risks involves the risk caused by dangerous substances, which spreads through the risk chain. When risks interact with other risk factors, a coupling effect occurs, leading to a change in the risk level, which may deviate from people’s expectations and lead to losses. The logistics risk of hazardous chemicals involves both internal factors and connections with the outside world. Generalized risk coupling includes the coupling within the system, as well as cross-coupling with external systems. The narrow sense of risk coupling refers to the mutual influence among various elements in the system. The typical coupling problem of hazardous chemical logistics risk is limited to the analysis of the hazardous chemicals logistics system, but does not consider the external factors of the hazardous chemicals transportation system. The risk connection of port hazardous chemicals logistics is an important link, and the interaction and influence of its risk factors are important factors leading to the emergence and development of risks. The influence of risk factors can be divided into single-, dual-, and multi-factor influence. Single risk refers to the interactive evolution of risk factors in each sub-system (e.g., personnel, machine, environment, or management), which is an important aspect affecting the evolution of the risk. The coupling of two elements involves two sub-systems that influence each other and are related to each other, which together affect the risk. Multi-factor coupling refers to the interaction of three or more risk factors.

Heterogeneous factor risk coupling refers to the coupling between two or more systems, including dual- and multi-factor coupling. In this paper, we take the dual-factor port hazardous chemical logistics risk as the research objective, including four sub-systems: personnel risk, ship risk, environmental risk, and management risk. Namely, the coupling effects between these four sub-systems are analyzed in six forms of risk coupling, including personnel–ship risk coupling, personnel–environmental risk coupling, personnel–management risk coupling, ship–environmental risk coupling, ship–management risk coupling, and environment–management risk coupling. The accidents caused by the coupling of these logistics risk factors are mostly harmful and concealed. From the perspective of the temporal and spatial evolution of risk, the interaction of multi-factor risk indicators in the coupled hierarchical network model indicates hierarchical transmission and sequential interaction based on the coupled chain. The port hazardous chemical logistics risk coupling hierarchical network model is shown in Figure 4 [30].

In the port hazardous chemical logistics risk system, considering the risk coupling relationship between personnel and ships, the operational behaviors of personnel may lead to uncertain risks in the logistics and transportation links. Unexpected situations in the logistics and transportation links can also affect the judgment ability of personnel, indicating a closed-loop relationship between them. Among them, when the professional skills of personnel do not meet the standard, poor discipline and unstable psychological quality will affect the logistics and transportation of hazardous chemicals. Strict requirements for the standardized operational process of hazardous chemicals can effectively prevent and control the accident risk in the logistics and transportation management of hazardous chemicals in ports. For example, in the closed loop of the personnel–ship risk system, based on the visual points of the personnel–ship and people-oriented relationships, the two make up for each other’s shortcomings and help to avoid accidents.

From a statistical point of view, the more coupled factors there are, the smaller the probability of this situation, which means that the probability of multi-factor coupling is much lower than that of two-factor coupling; however, when multi-factor coupling occurs, the losses that it may bring can be very large, even ten times or higher. When the personnel, ship, environmental, and management risk systems in the hazardous chemical logistics system are coupled, numerous accidents and new risks can arise. Under the comprehensive action of risk factors, such as those related to personnel, ships, environment, and management, the risk factors are increased, not only leading to property damage and casualties, but also potentially adversely affecting the working environment of the port, such as by increasing the explosion risk of dangerous chemicals. Figure 5 shows the interactions among personnel factors, ship factors, environmental factors, and management factors [31].

As shown in Figure 5, the coupling effects of the four risk factor systems also lay the foundation for the study of multi-factor coupling. In the personnel–ship–environmental–management risk coupling system, a root cause is the extreme weather problem in the environmental risk system, which strongly couples with the working environmental risk, in turn affecting the supervision and early warning risks in the management elements. The changes in supervision and early warning risks then affect the safety awareness of operators, where a lack of safety awareness of operators affects the ship’s loading state, resulting in heightened ship risk. However, a coupling approach based on the risk of management elements is as follows: the risk associated with personnel safety training in management elements affects the safety awareness of human factors, while a change in the safety concept changes the route condition risk in the environmental risk system; in turn, the route condition risk affects monitoring and early warning risks. In terms of the risk of ships, the maintenance and performance of transportation equipment is affected, thus forming a coupling system of personnel–ship–environment–management risks, with management factors as the triggering source.

### 2.3. Port Hazardous Chemical Logistics Risk Accident Cause Mechanism

In the port hazardous chemical logistics risk system, the coupling effect of a single factor rarely leads to serious safety accidents, due to the influence of the self-regulating resources of the logistics system. In the case of single-factor risk or coupling, the defense mechanism in the logistics chain can stop the risk and prevent it from reaching a critical point, thus realizing zero coupling and/or negative coupling. Personnel, ship, environmental, and management systems in the port chemical logistics system have different degrees of defects or errors. After breaking through their own protection systems, they may run safely along the logistics system. When other factors arise, these risks may spread at an extremely fast speed, and the critical point of risks are altered through risk coupling. If the critical point of a risk exceeds the critical value, the negative coupling of risk systems may occur, increasing the occurrence of accidents and even generating new risks. The formation mechanism of the port hazardous chemical logistics risk system is depicted in Figure 6 [32].

## 3. Application of Port Hazardous Chemical Logistics Risk Coupling Model

### 3.1. Build an N-K Model of Port Hazardous Chemicals Logistics Risk Coupling

The N-K model was put forward by Kauffman in 1993, inspired by the combinatorial evolution of biological genes. It mainly contains two important parameters, N and K. N represents the total number of components in the system. If there are N components in the system, and each component has N states, then all combinations exist. K represents the number of interdependent relationships among components. The minimum value of K is zero, and the maximum value of K is one When K reaches a certain level, the relationships among components can form a network.

The basic principle of using the N-K model to analyze the risk coupling of hazardous chemical logistics is to measure the impact of coupling on the risk system of hazardous chemical logistics by calculating the interactive information value *T* among the risks associated with personnel, mechanical equipment, the environment, management, and other factors [33]. The larger the value of *T*, the more times that these factors are coupled in some way and the greater the risk of accidents. The formula for the coupling risk T is:(1)$$T\left(a,b,c,d\right)=\sum_{h}^{H}\sum_{i}^{I}\sum_{j}^{J}\sum_{k}^{K}P_{h,i,j,k}\times \log_{2}\left(P_{h,i,j,k}/\left(P_{h\ldots .}P_{.i..}P_{..j.}P_{\ldots k}\right)\right)$$
where *a*,*b*,*c*,*d* represent the four considered risk factors (e.g., personnel, ship, environmental, and management risk); $P_{h,i,j,k}$ is the probability of coupling between personnel in state *h*, ship in state *i*, environment in state *j*, and management in state k(h=1,…,H;i=1,…,I;j=1,…,J;k=1,…,K); and $P_{h\ldots }P_{.i..}P_{..j.}P_{\ldots k}$ denotes that the personnel, ship, environment, and management are in states h,i,j,k, respectively.

Single-factor risk coupling refers to the interactions between single factors that affect the logistics safety of hazardous chemicals in ports. There are four kinds of single-factor coupling; namely, personnel factor coupling, ship factor coupling, environmental factor coupling, and management factor coupling, respectively denoted as $T_{11(a)}$, $T_{12(b)}$, $T_{13(c)}$, and $T_{14(d)}$. The total value of risk factor coupling is denoted by $T_{1}$.

Two-factor coupling risk refers to the risk caused by the interaction and influence of two risk factors that affect the logistics risk of hazardous chemicals in port, including personnel–ship coupling risk, personnel–environmental coupling risk, personnel–management coupling risk, ship–environmental coupling risk, ship–management coupling risk, and environment–management coupling risk, denoted as $T_{21}\left(a,b\right)$, $T_{22}\left(a,c\right)$, $T_{23}\left(a,d\right)$, $T_{24}\left(b,c\right)$, $T_{25}\left(b,d\right)$, and $T_{26}\left(c,d\right)$, respectively. These are calculated as follows:(2)$$T_{21}\left(a,b\right)=\sum_{h=1}^{H}\sum_{i=1}^{I}P_{h,i}\times \log_{2}\left(P_{h,i}/\left(P_{h\cdots }P_{.i..}\right)\right)$$
(3)$$T_{22}\left(a,c\right)=\sum_{h=1}^{H}\sum_{j=1}^{J}P_{h,j}\times \log_{2}\left(P_{h,j}/\left(P_{h\cdots }P_{.j.}\right)\right)$$
(4)$$T_{23}\left(a,d\right)=\sum_{h=1}^{H}\sum_{k=1}^{K}P_{h,k}\times \log_{2}\left(P_{h,k}/\left(P_{h\cdots }P_{..k}\right)\right)$$
(5)$$T_{24}\left(b,c\right)=\sum_{i=1}^{I}\sum_{j=1}^{J}P_{i,j}\times \log_{2}\left(P_{i,j}/\left(P_{.i..}P_{..j.}\right)\right)$$
(6)$$T_{25}\left(b,d\right)=\sum_{i=1}^{I}\sum_{k=1}^{K}P_{i,k}\times \log_{2}\left(P_{i,k}/\left(P_{.i..}P_{\ldots k}\right)\right)$$
(7)$$T_{26}\left(c,d\right)=\sum_{j=1}^{J}\sum_{k=1}^{K}P_{j,k}\times \log_{2}\left(P_{j,k}/\left(P_{..j.}P_{\ldots k}\right)\right)$$

Multi-factor coupling refers to the risk coupling caused by the interaction of three or more risk factors. The three-factor coupling risks in the port hazardous chemicals logistics risk system include the coupling risks of personnel–ship–environment, personnel–ship–management, personnel–environment–management, and ship–environment–management, denoted as $T_{31}\left(a,b,c\right)$, $T_{32}\left(a,b,d\right)$, $T_{33}\left(a,c,d\right)$, and $T_{34}\left(b,c,d\right)$, respectively. These values are calculated as follows:(8)$$T\left(a,b,c\right)=\sum_{h=1}^{H}\sum_{i=1}^{I}\sum_{j=1}^{J}P_{h,i,j}\times \log_{2}\left(P_{h,i,j}/\left(P_{h\ldots }P_{..i.}P_{..j.}\right)\right)$$
(9)$$T\left(a,b,d\right)=\sum_{h=1}^{H}\sum_{i=1}^{I}\sum_{k=1}^{K}P_{h,i,k}\times \log_{2}\left(P_{h,i,k}/\left(P_{h\ldots }P_{..i.}P_{\ldots k}\right)\right)$$
(10)$$T\left(a,c,d\right)=\sum_{h=1}^{H}\sum_{j=1}^{J}\sum_{k=1}^{K}P_{h,j,k}\times \log_{2}\left(P_{h,j,k}/\left(P_{h\ldots }P_{..j.}P_{\ldots k}\right)\right)$$
(11)$$T\left(b,c,d\right)=\sum_{i=1}^{I}\sum_{j=1}^{J}\sum_{k=1}^{K}P_{i,j,k}\times \log_{2}\left(P_{i,j,k}/\left(P_{..i.}P_{..j.}P_{\ldots k}\right)\right)$$

### 3.2. Determination of Probability and Coupling Value of Logistics Risk Factors of Hazardous Chemicals in Port

Taking Tianjin Port as an example, we collected effective information and data through a field investigation and expert consultation, then analyzed the change in the coupling degree of three factors in the four systems of personnel, ship, environment, and management from January to October of 2021. We used the N-K model to further study the coupling effects of the three factors in the port hazardous chemical logistics risk system.

By collecting and sorting relevant data on Tianjin Port, the probabilities of risk events caused by single-factor coupling, double-factor coupling, and multi-factor coupling were obtained, and the mean value, deviation degree, and fuzzy probability were further obtained by calculating Formulas (2)–(11). The results are provided in Table 1 below.

According to Formulas (2)–(7), the risk coupling value under the coupling effect of different risk factors was calculated, as given in Table 2.

In Table 2, $P_{0\ldots }$ is the probability of risk occurrence when personnel factors are not involved in coupling, while $P_{1\ldots }$ is the probability of risk occurrence when personnel factors are involved in coupling, “.” stands for zero or one, and $P_{0\ldots }$ = $P_{0000}$ + $P_{0100}$ + $P_{0010}$ + $P_{0001}$ + $P_{0110}$ + $P_{0101}$ + $P_{0011}$ + $P_{0111}$ = 0.4825. The meaning and calculation results for the other single-factor risk probabilities in Table 2 can be determined similarly.

The two-factor risk probabilities of port hazardous chemical logistics risk can be obtained by determining the two-factor coupling values, as shown in Table 3.

In Table 3, $P_{00..}$ is the probability of risk occurrence when personnel and ship factors are not involved in the coupling, while $P_{01..}$ is the probability of risk occurrence when personnel factors are not involved and ship factors are involved in the coupling, “.”stands for zero or one, and $P_{00..}$ = $P_{0000}$ + $P_{0010}$ + $P_{0001}$ +$P_{0011}$ = 0.129. The meanings and calculation results for the other two-factor risk probabilities in Table 3 can be obtained similarly.

In Table 4, $P_{000.}$ denotes the probability of risk occurrence when personnel, ship, and environmental factors are not involved in the coupling, where “.”stands for zero or one. Here, $P_{000.}$ = $P_{0000}$ + $P_{0001}$ = 0.0687. The meanings and calculation results of risk probability for the other three-factor combinations in Table 4 can be obtained similarly.

## 4. Simulation Study on the Coupling Effect of Port Hazardous Chemical Logistics Risk Based on System Dynamics

Taking Tianjin Port as an example, we analyzed the coupling effect of hazardous chemical logistics risk factors according to the obtained port hazardous chemical logistics risk data. According to the causes of risk accidents, the risk factors were divided into four sub-systems: personnel, ship, environment, and management. The coupling effect within the same sub-system is considered as homogeneous factor risk coupling. Meanwhile, heterogeneous factor risk coupling refers to the coupling effects among multiple sub-systems; that is, the coupling effects among multiple risk factors of multiple sub-systems among the four considered risk sub-systems [34].

### 4.1. Construction of an N-K Model of Port Hazardous Chemical Logistics Risk Coupling

According to the results of Formulas (2)–(7) and Table 4, the two-factor coupling degree in port hazardous chemical logistics risk was calculated, as shown in Table 5.

By sorting the two-factor coupling values, we can obtain the change in the two-factor coupling degree for the port hazardous chemical logistics risk system.

From Figure 7, we can draw the following conclusions: the values of certain risk systems—namely, personnel–ship, personnel–environment, and personnel–management systems—were relatively large and, so, these three risk systems were strongly coupled. It is not difficult to see that all three risk systems contain personnel factors, which easily produce coupling, resulting in the maximum coupling value of the risk systems being related to human beings. The ship–management risk system presented a moderate coupling effect, indicating that the associated coupling effect was not very strong, but was at a moderate level, such that it is necessary to strengthen management and control in this aspect. The ship–environment and environment–management risk systems presented a low coupling state and the coupling effect was weak, such that enterprises only need to take precautions and pay attention to them in time.

We next constructed the two-factor flow chart and variable set, as shown in Table 6 and Figure 8 [35].

According to the coupling situation, a system dynamics risk coupling model was constructed to simulate the personnel, ship, environmental, and management risk systems. The constructed model is shown in Figure 8.

Selecting the risk coupling data of four systems (i.e., personnel, ship, environment, and management) in Tianjin Port from January to December 2021, the comprehensive risk value and the change trend of the coupling degree for the four systems were obtained through simulation using the Vensim PLE software 5.9c [36].

As shown in Figure 9, in the port hazardous chemicals logistics risk, it can be found that the personnel risk system and ships were on the rise. The environmental risk system and the management risk system were in a stable operation state at first, but they were in a downward trend with the accumulation of time, and the risk coupling value gradually decreased with the passage of time. The ship risk value decreased at the turning point in May. A management risk system was emerging. The longer the fluctuation of personnel risk system rises, the stronger the risk coupling effect; the cliff-like change started from July, rose rapidly in November, and dropped to an extreme value from October to November. On the whole, the ship risk showed an upward trend, but there was an inflection point in September, showing an upward trend. At first, the environmental risk system was in a high value state, but it turned into a declining state in October. The risk management system was running smoothly and showed a downward trend in September.

Through analysis of the two-factor risk coupling, we found that the two-factor coupling of port hazardous chemical logistics risks developed slowly in the initial stage, but the coupling effect tended to accelerate with the passage of time. However, in the actual transportation of hazardous chemicals in ports, the risk does not only involve the risk coupling of two risk systems. Above, we analyzed the dual-factor coupling based on the personnel–ship–environment–management risk system. Next, we simulated the coupling effect of three factors in this risk system in order to better explain the coupling effects of the four risk sub-systems. By analyzing the coupling effect of port hazardous chemical logistics risk under two and three factors, we could more systematically and comprehensively analyze the mechanism of the coupling effect of port hazardous chemical logistics risk, allowing for deeper research on the coupling effect of port hazardous chemical logistics risk.

### 4.2. Simulation of Three-Factor Coupling in the Logistics Risk of Port Hazardous Chemicals Based on System Dynamics

#### 4.2.1. Constructing the Three-Factor Coupling Index System and Simulation Model

According to the results of Formulas (8)–(11) and Table 4, the coupling values of port hazardous chemical logistics risk factors were calculated, as shown in Table 7.

From Figure 10, we can draw the following conclusions: The value of the personnel–ship–management risk system indicates a high-risk coupling state, which shows that the personnel–ship–management risk system is characterized by strong coupling. It is not difficult to see that both the personnel–ship–environment and personnel–environment–management risk systems contain human factors and environmental factors, both of which are easily coupled with other factors, thus leading to the general coupling of risks related to people. The ship–environment–management risk system presented a low coupling effect, indicating that its coupling effect was not very strong and was at a low level, so it is only necessary to strengthen control.

It can be seen that the personnel risk factors occupied the main position in the port hazardous chemical logistics safety accidents, while ship risk factors also maintained a very important position. Therefore, in order to reduce the occurrence of port hazardous chemical logistics accidents, attention should be paid to the internal coupling state of personnel factors and other sub-system factors, as well as the evolving trend between them, in order to prevent the coupling between personnel risk factors and ship, environment, and management risk factors, thus reducing the harm degree of accidents [37].

We constructed a three-factor coupling index system for the port hazardous chemical logistics risk system, as detailed in Table 8.

The flow chart establishing the causality of system dynamics was obtained using Vensim PLE software 5.9c, which is shown in Figure 11. By bringing a set initial value into the equation, we could intuitively track the risk level change trends in the personnel–ship–management risk coupling system with the same software, as shown in Figure 12.

As shown in Figure 12, the total risk level of the personnel–ship–management coupling system, regarding the port hazardous chemical logistics risk, remained in the assigned state at first, maintaining a stable value from the first month to the fifth month. Then, a rapid increase can be observed from the sixth month. Therefore, timely management and control should be conducted before the coupling risk occurs, in order to avoid its rapid growth and reduce the coupling risk to a minimum. When the risk formation presents slow growth, intervention should be conducted to reduce the speed of coupling risk formation and avoid uncontrolled risk accidents due to strong coupling of the risk system [38].

#### 4.2.2. Simulation Research on Personnel–Ship–Management Risk System

Changing the coupling coefficients based on the personnel risk system.

We selected two risk coupling coefficients related to personnel risk, ship risk, and management risk, respectively, and observed the change in risk level of the personnel–ship–management risk system when changing the coupling degree. Coupling coefficient C3 (between equipment failure, aging wear, and illegal operation of equipment and facilities) and coupling coefficient C4 (between knowledge and quality of operators, safety training, and safety education and education management) were selected as the variable coupling coefficients. Based on their original values, two schemes were adopted (i.e., increased by 30% and decreased by 30%), as detailed in Table 9 [39].

As shown in Figure 13, by changing the coupling coefficient C3 (between equipment failure, aging wear of equipment and facilities, and illegal operation) and the coupling coefficient C4 (between knowledge and quality of operators, safety training of personnel, safety education, and education management), when the coupling coefficient was reduced by 30%, the growth rate of the trend line was obviously lower than that of the initial value, indicating that timely management and control of risk coupling can effectively reduce the occurrence of risks. When the coupling coefficient increased by 30%, we intuitively observed that the growth showed a rapid increase in stages until August, following which the risk level became larger and the growth rate increased. With the increase in the risk coupling coefficient, the risk generation speed becomes faster and the control difficulty becomes greater; when the coupling coefficient becomes smaller, the coupling effect is weak, and the risk value is also small.

Changing the coupling coefficients of the ship risk system.

We selected two risk coupling coefficients related to ship risk, personnel risk, and management risk, respectively, and observed the change in risk level of the personnel–ship–management risk system by changing the coupling degree. Here, coupling coefficient C1 (between operator’s knowledge and quality, illegal operation, and equipment failure), and coupling coefficient C2 (between the characteristics of hazardous chemicals, the ship’s loading state, and the establishment of a safety system) were selected as the variable coupling coefficients. Based on the original numerical values, two schemes were adopted (increased by 30% and decreased by 30%), as detailed in Table 10.

Figure 14 shows the result when changing coupling coefficient C1 (between the knowledge and quality of operators, illegal operation, and equipment failure) and coupling coefficient C2 (between the characteristics of hazardous chemicals, the ship’s loading state, and safety system). When the coupling coefficient increased by 30%, the growth curve showed a high-speed increase, especially after September, when the risk coupling effect reached its maximum and irreparable accidents could occur if control was not strengthened. When the coupling coefficient decreased by 30%, we can see that, although the growth rate fluctuated, it remained close to zero before August and the growth rate was low, indicating that the risk coupling had been effectively controlled at this time.

Changing the coupling coefficients of the personnel–ship–management risk system.

Next, we selected risk coupling factors related to the personnel–ship–management risk system and observed the impact of changing the coupling coefficient on the total risk level of the personnel–ship–management system. Coupling coefficient C1 (between operator’s knowledge and quality, illegal operation, and equipment failure), coupling coefficient C6 (between safety awareness, safety system establishment, and the characteristics of hazardous chemicals), and coupling coefficient C7 (between ship’s load, monitoring and early warning investment, and safety education and education management) were selected as the variable coupling coefficients. Based on the initial values, two schemes were adopted (increased by 30% and decreased by 30%), as detailed in Table 11.

As shown in Figure 15, when decreasing coupling coefficient C1 (between knowledge and quality of operators, illegal operation, and equipment failure), coupling coefficient C6 (between safety awareness, safety system establishment, and the characteristics of hazardous chemicals), and coupling coefficient C7 (between ship load, monitoring and early warning investment, and safety education and education management) were decreased by 30% at the same time, the increase in the growth curve was gentle and the curve showed a negative growth trend before July. Until the tenth month the growth was under control, at which time the coupling risk should be under the control of the enterprise(s). When the coupling coefficients were increased by 30%, we can see that the accumulated risk level changed gradually with time and presented a state of rapid growth, making it uncontrollable for the enterprise(s). Therefore, before the risk coupling effect is strengthened, timely management and control is required to avoid the occurrence of safety accidents.

### 4.3. Decoupling Principle of Risk Coupling in Port Hazardous Chemical Logistics

The concept of coupling has been widely used in communication technology, computer technology, mathematical models, and so on. However, the risk coupling of port hazardous chemical logistics is irreversible, and cannot be restored to the original level. Therefore, the decoupling of port hazardous chemical logistics involves the interaction between risk prevention and control elements (see Figure 16 for a depiction of the decoupling mechanism of port hazardous chemical logistics risk). The existence of risk coupling speeds up the risk occurrence rate; however, it also increases the risk, which is a hidden risk. Therefore, we need to utilize the decoupling principle to reduce the coupling of risks, thus changing the direction of coupling to reduce the risks inherent to the logistics and transportation of hazardous chemicals in ports [40].

When the risk material flows in the risk chain, if the risk factors meet the segregator and are in the control area of the segregator before the coupling action starts, the occurrence of risk A will be reduced, while the risk degree of risk A will be reduced if it is outside the control area of the segregator. Similarly, the risk degree of risk B will also be reduced. Outside the scope of the segregator, the coupling between various factors is inevitable. Thus, we may adopt an effective method to adjust the coupling degree between A and B; that is, when the risk of A is at its peak, we should reduce the risk of B to its lowest point as much as possible, thus reducing the coupling effect between A and B. Next, risks A and B meet at the peak, and both risks meet at the beginning of the peak. The coupling effect of the two risks will produce a strong alliance, leading to huge risks that cannot be controlled. Therefore, we can add a segregator between them, such that the risks of A and B can be prevented from converging at the peak and then converging at the lowest point. By transferring the positive coupling of risks to negative coupling or zero coupling, the coupling intensity can be greatly reduced, thus providing valuable time for enterprises to deal with risks in the risk environment [41,42].

### 4.4. Suggestions for Prevention and Control of Port Hazardous Chemical Logistics Risk Coupling System

Single risk source of control sub-system

In the risk system of hazardous chemical logistics, it is necessary for hazardous chemical logistics enterprises to identify the existing risks before paired sub-systems of heterogeneous factors are coupled, thus fundamentally eliminating the coupling effect. At the same time, the risks associated with each sub-system should be monitored and controlled in a timely manner, the changes of transportation equipment and environment should be monitored, possible risks should be discovered and identified over time, and a corresponding risk management system should be formulated. Through the dynamic changes of different risk sources, the risk value can be adjusted and the corresponding control priority may be formulated, thus ensuring the accurate control of risk sources and prevention of accidents. At the same time, we should strengthen the management and control of risk sources.

Stagger the peak of sub-system risk

First of all, we need to determine when the peak sub-system risk will occur, such that we can stagger the peaks in time. Foremost, it is necessary to understand the risk information of the enterprise. In the process of transporting dangerous goods, we must make full use of the risk information of dangerous chemicals, carry out effective identification, and ensure safe information transmission. Secondly, we should strengthen the independence of functional departments and posts. If our management department can minimize the peak risk in the shortest time, then there will be no risk. In the case of high-risk human factors, we should strengthen the control and monitoring of human factors. This may prevent the collision between personnel risk and other risk factors, thus reducing the occurrence of dangerous accidents.

Avoid the vulnerability of coupled risk sub-systems.

The hazardous chemical logistics risk system is a typical fragile system, where its fragile characteristics will manifest in the whole logistics process of hazardous chemicals. Furthermore, the complexity of the hazardous chemical logistics system makes it especially vulnerable. Risks tend to be concentrated in relatively weak areas; that is, the areas that are the most vulnerable are the most prone to risks. Therefore, at this stage, we should adopt the coupled flow of risk constraints to transfer the vulnerable risks to areas with higher risks.

## 5. Conclusions

The causes of port hazardous chemical logistics risk are complex and inter-related [43]. We introduced complex system theory to study the causes of hazardous chemical logistics risk, which helped to obtain the basic law of the mutual influence of risk factors. It can be seen that the coupling effects in the hazardous chemical logistics risk system of a port not only influence each other, but are characterized by various strong and weak coupling effects.Based on the N-K model, coupling models of port hazardous chemical logistics risk under system dynamics were constructed [33]. Through quantitative analysis of the coupling relationships among sub-systems, it was found that the coupling value of the personnel–ship system was the strongest when considering the dual-factor coupling effect. Meanwhile, under the three-factor coupling effect, the coupling value of the personnel–ship–management risk system was the largest and the associated coupling effect was the strongest.In a coupled system, if many sub-systems are present, the coupling effect is greater and the risk fluctuation is smaller; however, the coupling risk caused by the surrounding objective factors is more difficult to control than that associated with subjective factors. Therefore, the objective factors cannot be ignored in the research of port hazardous chemical logistics risk.Two-factor and three-factor risk sub-systems were simulated. Through the simulation study on the coupling of two factors, according to the risk change, the result of interaction between the influencing factors—that is, the linkage of the influencing factors—increased the degree of risk. In brief, the greater the coupling force, the higher the degree of risk. The increase in risk degree caused by multi-factor coupling was greater than that caused by dual factors. System dynamics simulation of the three-factor risk coupling in the personnel–ship–management system was also carried out, and the influence of changes in different factors on the risk system was analyzed. Finally, we concluded that three-factor risk coupling is extremely fast and difficult to control; as such, it is necessary to intervene and control risks in a timely manner, before the coupling is strengthened.

The results show that the risk coupling effect between the personnel risk system and other systems is the most obvious in the port hazardous chemicals logistics risk system, and enterprises need to strengthen the management of personnel, find the risk sources as early as possible, control the risks in time, and reduce the severity of accidents. The research in this paper provides a new research perspective for the safe operation of port hazardous chemicals logistics, escorts the safe transportation of logistics industry and reduces the occurrence of hazardous chemicals logistics safety accidents. This paper analyzes the risk coupling effect of port hazardous chemicals logistics, which provides experience for the safety management of port enterprises in the future. However, this paper mainly focuses on the whole link of port logistics and transportation, and can pay attention to the risk coupling effect of other links of port hazardous chemicals, which promotes the development of port safety management.

## Figures and Tables

**Figure 1 ijerph-20-04008-f001:**
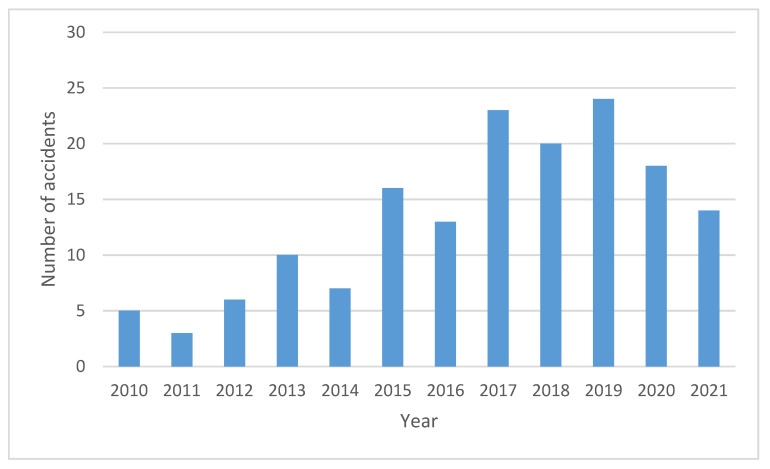
Statistics of port hazardous chemicals safety accidents.

**Figure 2 ijerph-20-04008-f002:**
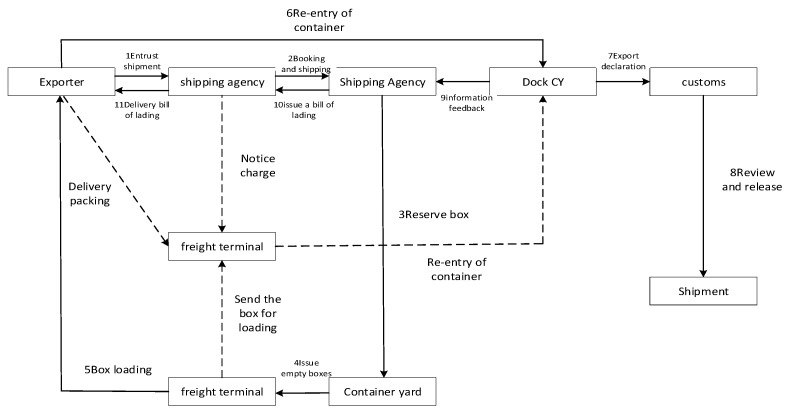
Flow chart of port hazardous chemicals logistics loading and unloading.

**Figure 3 ijerph-20-04008-f003:**
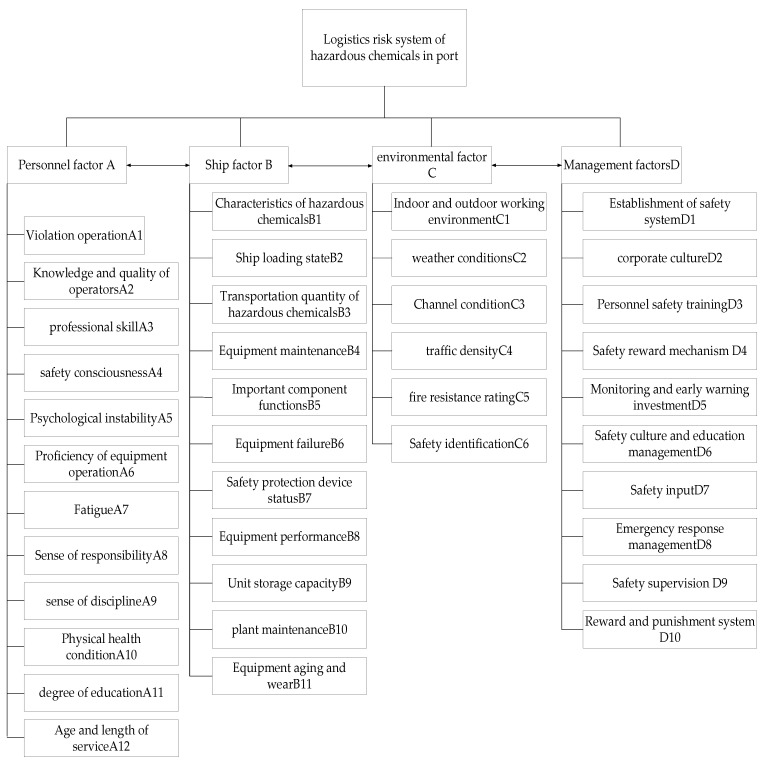
Flow chart of port hazardous chemicals logistics risk system.

**Figure 4 ijerph-20-04008-f004:**
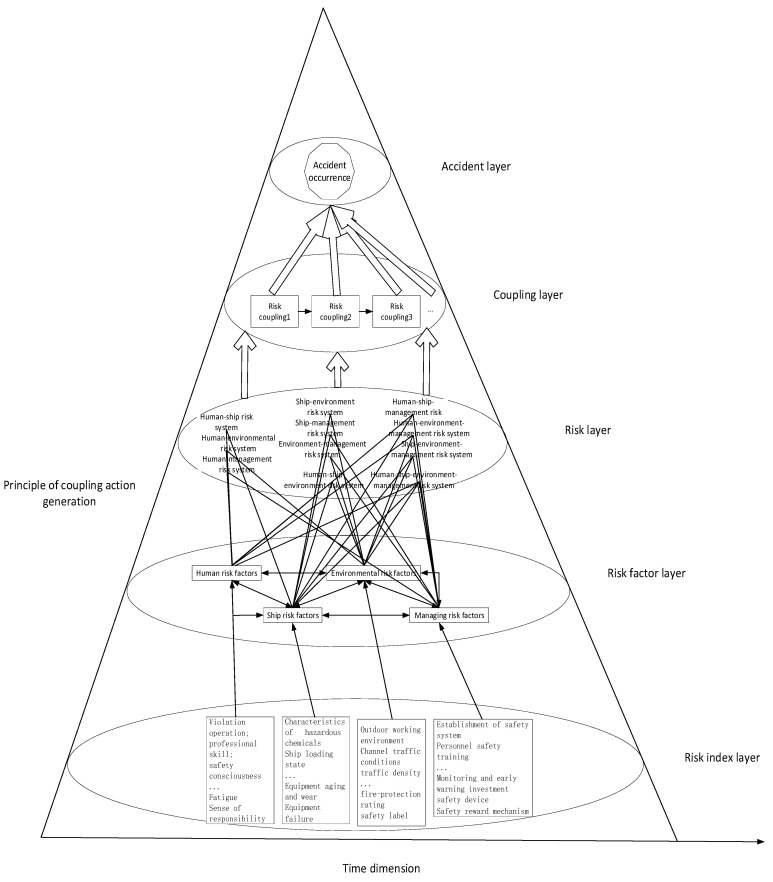
Port hazardous chemical logistics risk coupling hierarchical network model.

**Figure 5 ijerph-20-04008-f005:**
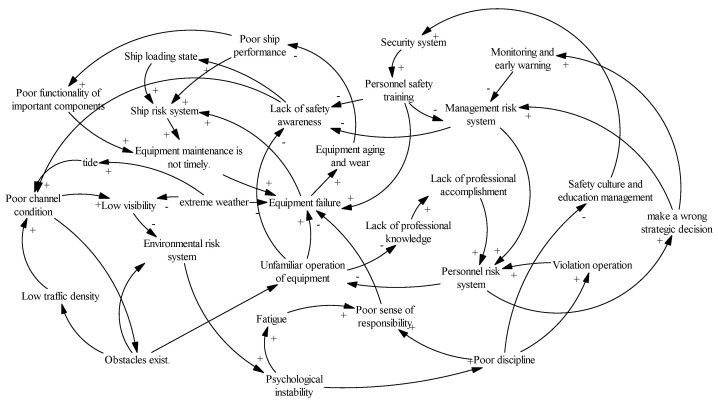
Risk coupling relationships between personnel, ship, environmental, and management factors (+ means positive feedback, - means negative feedback).

**Figure 6 ijerph-20-04008-f006:**
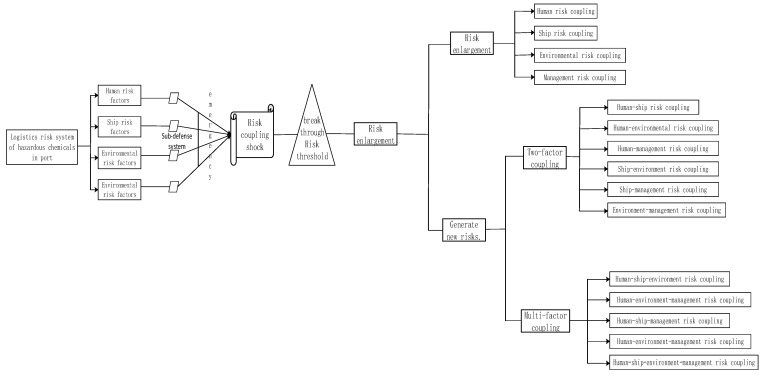
Formation mechanism of positive coupling in port hazardous chemical logistics risk system.

**Figure 7 ijerph-20-04008-f007:**
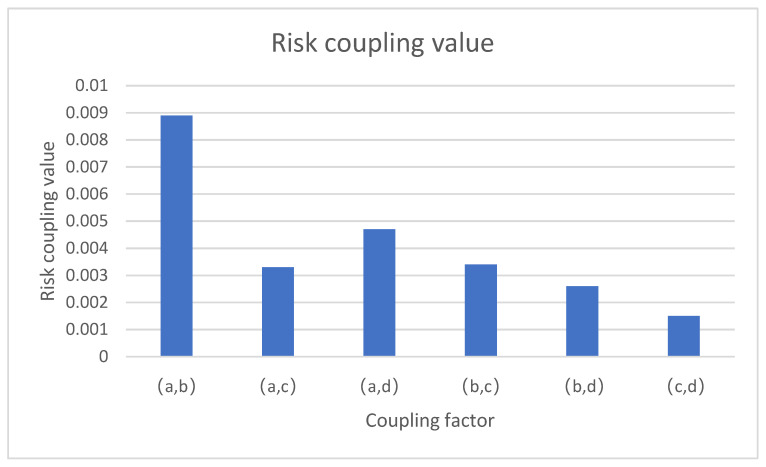
Two-factor risk coupling values for port hazardous chemical logistics risk.

**Figure 8 ijerph-20-04008-f008:**
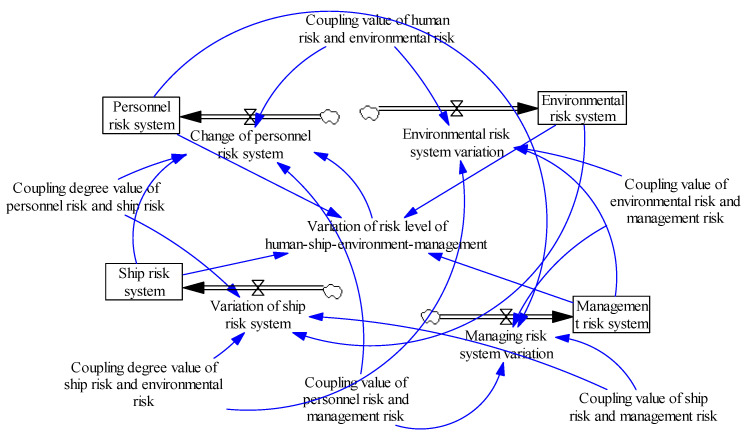
Two-factor coupling model for port hazardous chemical logistics risk system.

**Figure 9 ijerph-20-04008-f009:**
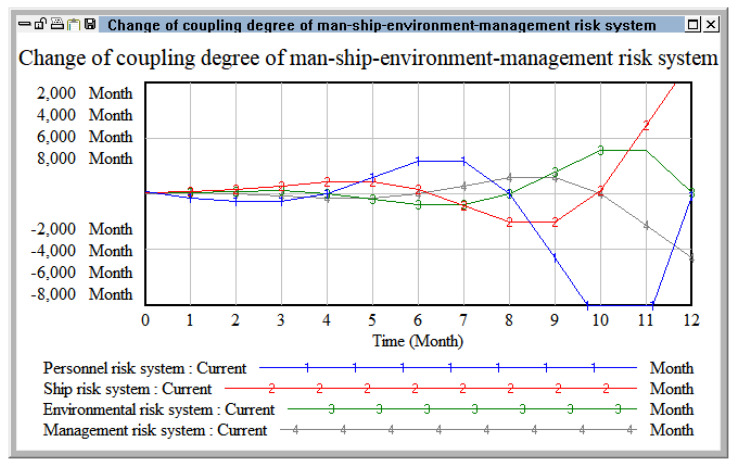
Change in coupling degree for personnel-ship-environment-management risk system.

**Figure 10 ijerph-20-04008-f010:**
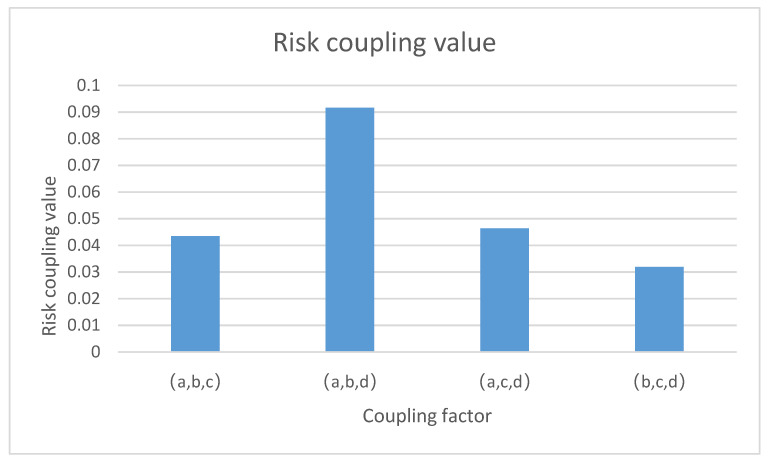
Three-factor risk coupling values for port hazardous chemical logistics risk.

**Figure 11 ijerph-20-04008-f011:**
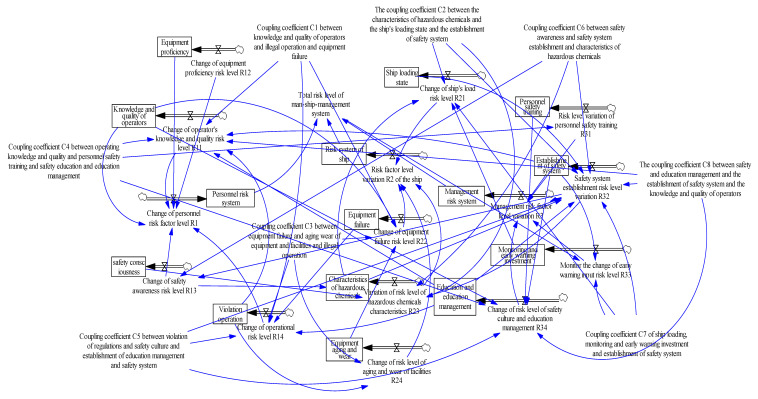
Three-factor coupling relationships in port hazardous chemical logistics risk.

**Figure 12 ijerph-20-04008-f012:**
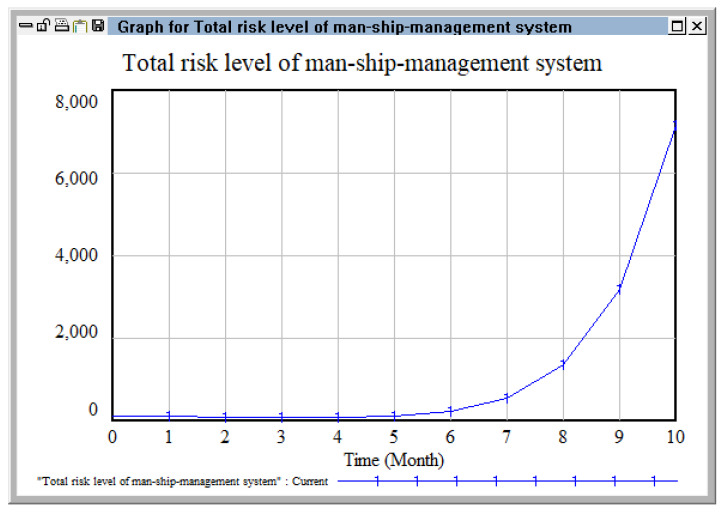
Change of risk level in personnel–ship–management system.

**Figure 13 ijerph-20-04008-f013:**
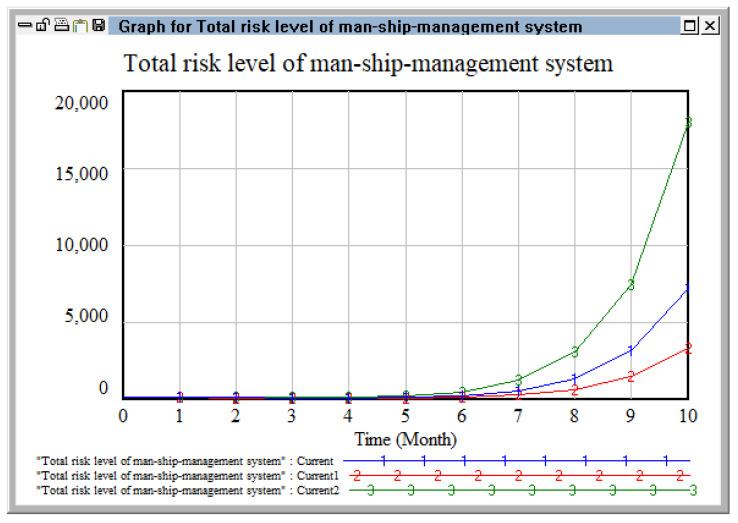
Change of risk level of man ship management system by changing coupling coefficient.

**Figure 14 ijerph-20-04008-f014:**
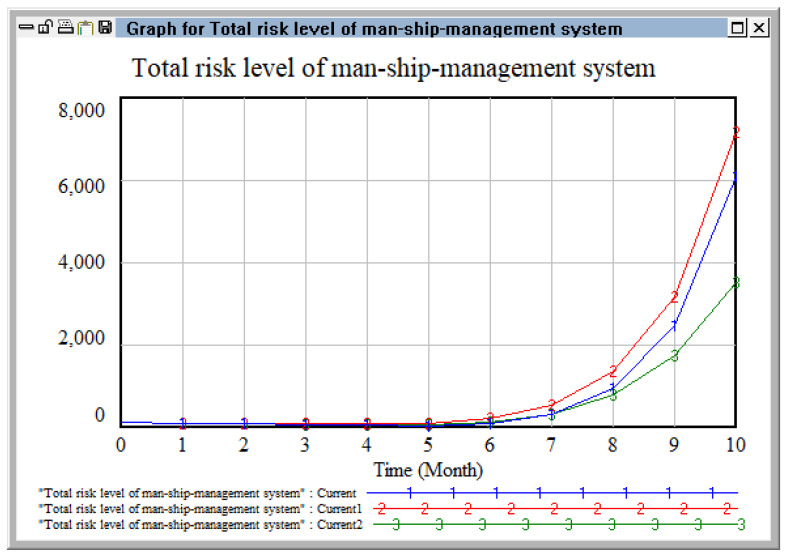
Changes in the risk level of the personnel–ship–management system under the two schemes.

**Figure 15 ijerph-20-04008-f015:**
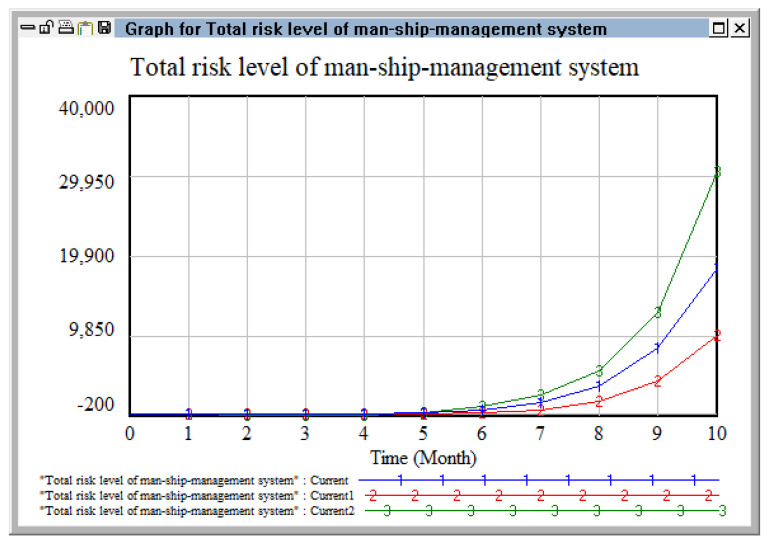
Change in the risk level of the personnel–ship–management system under the two schemes.

**Figure 16 ijerph-20-04008-f016:**
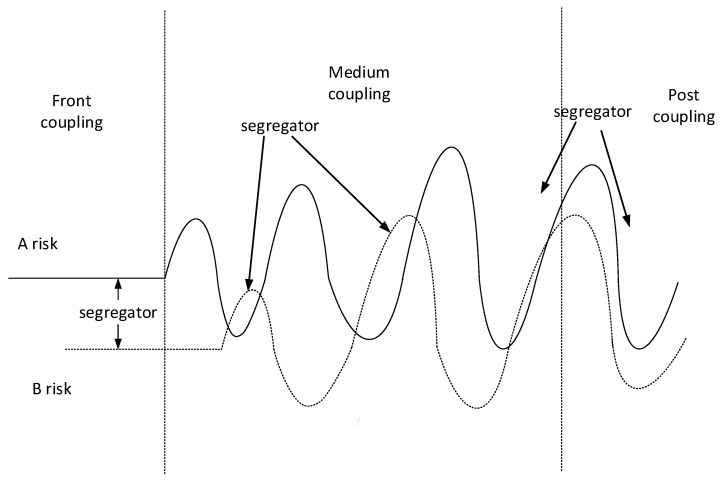
Decoupling principle of hazardous chemicals logistics risk.

**Table 1 ijerph-20-04008-t001:** Probability of coupling events of port hazardous chemicals logistics risk factors.

Single-Factor Coupling	Two-Factor Coupling	Three-Factor Coupling
$P_{0000}$ = 0	$P_{1100}$ = 0.0841	$P_{1110}$ = 0.0329
$P_{1000}$ = 0.0541	$P_{1010}$ = 0.0499	$P_{1101}$ = 0.0817
$P_{0100}$ = 0.0452	$P_{1001}$ = 0.0318	$P_{1011}$ = 0.0897
$P_{0010}$ = 0.0447	$P_{0110}$ = 0.0788	$P_{0111}$ = 0.1414
$P_{0001}$ = 0.0687	$P_{0101}$ = 0.0881	$P_{1111}$ = 0.0933
	$P_{0011}$ = 0.0156	

**Table 2 ijerph-20-04008-t002:** Single-factor risk probability of port hazardous chemicals logistics risk.

Single Factor Risk Rate	
$P_{0\ldots }$ = 0.4825	$P_{..1.}$ = 0.5463
$P_{1\ldots }$ = 0.5175	$P_{\ldots 0}$ = 0.3897
$P_{.1..}$ = 0.6455	$P_{\ldots 1}$ = 0.6103
$P_{..0.}$ = 0.4537	$P_{.0..}$ = 0.3545

**Table 3 ijerph-20-04008-t003:** Two-factor risk probability in port hazardous chemical logistics.

	Two-Factor Coupling	
$P_{00..}$ = 0.129	$P_{0.0.}$ = 0.1687	$P_{0..1}$ = 0.3138
$P_{01..}$ = 0.3535	$P_{1..0}$ = 0.221	$P_{.0.0}$ = 0.1487
$P_{10..}$ = 0.2255	$P_{1..1}$ = 0.2965	$P_{.1.0}$ = 0.241
$P_{11..}$ = 0.292	$P_{..01}$ = 0.2703	$P_{.0.1}$ = 0.2058
$P_{0.0.}$ = 0.202	$P_{.00.}$ = 0.1546	$P_{.1.1}$ = 0.4045
$P_{1.0.}$ = 0.2517	$P_{.10.}$ = 0.2991	$P_{..00}$ = 0.1834
$P_{0.1.}$ = 0.2805	$P_{.01.}$ = 0.1999	$P_{..10}$ = 0.2063
$P_{1.1.}$ = 0.2658	$P_{.11.}$ = 0.3464	$P_{..11}$ = 0.34

**Table 4 ijerph-20-04008-t004:** Three-factor risk probabilities in port hazardous chemical logistics.

	Three-Factor Risk Rate	
$P_{000.}$ = 0.0687	$P_{.010}$ = 0.0946	$P_{0.00}$ = 0.0452
$P_{100.}$ = 0.0859	$P_{.001}$ = 0.1005	$P_{1.00}$ = 0.1382
$P_{010.}$ = 0.1333	$P_{00.0}$ = 0.0447	$P_{0.10}$ = 0.1235
$P_{001.}$ = 0.0603	$P_{10.0}$ = 0.194	$P_{0.01}$ = 0.1568
$P_{110.}$ = 0.1658	$P_{01.0}$ = 0.124	$P_{1.10}$ = 0.0828
$P_{101.}$ = 0.1396	$P_{00.1}$ = 0.0843	$P_{1.01}$ = 0.1135
$P_{011.}$ = 0.2202	$P_{.110}$ = 0.1117	$P_{0.11}$ = 0.157
$P_{111.}$ = 0.0329	$P_{.101}$ = 0.1698	$P_{1.11}$ = 0.183
$P_{.000}$ = 0.0541	$P_{.011}$ = 0.1053	$P_{11.0}$ = 0.117
$P_{.100}$ = 0.1293	$P_{.111}$ = 0.2347	$P_{10.1}$ = 0.1215
$P_{01.1}$ = 0.2295	$P_{11.1}$ = 0.175	

**Table 5 ijerph-20-04008-t005:** Two-factor coupling values of port hazardous chemical logistics risk.

Coupling Factor	Risk Coupling Value
$T_{2}^{1}$ (a,b)	0.0089
$T_{2}^{2}$ (a,c)	0.0033
$T_{2}^{3}$ (a,d)	0.0047
$T_{2}^{4}$ (b,c)	0.0034
$T_{2}^{5}$ (b,d)	0.0026
$T_{2}^{6}$ (c,d)	0.0015

**Table 6 ijerph-20-04008-t006:** System Flow Rate and Variable Set of Person-Ship.

Variable Name		Actual Meaning of Variable
Horizontal variable	L1 (t)	Risk level indicator of risk behavior of the personnel sub-system.
	L2 (t)	Indicates the risk level index of the ship risk sub-system.
	L3 (t)	Indicates the risk level index of the environmental risk sub-system.
	L4 (t)	Indicates the risk level indicator of the management risk sub-system.
Auxiliary variable	RCHGFXSP	The index of increase per unit time indicating the risk level of man, ship, environment, or management.
Rate variable	R1	The increase in the behavioral risk level of personnel per unit time.
	R2	The increase in the risk level of the ship per unit time.
	R3	The increase in environmental risk level per unit time.
	R4	The increase in management risk level per unit time.
Constant	C1	Coupling coefficient of personnel risk and ship risk.
	C2	Coupling coefficient of personnel risk and environmental risk.
	C3	Coupling coefficient of personnel risk and management risk.
	C4	Coupling coefficient of ship risk and environmental risk.
	C5	Coupling coefficient of ship risk and management risk.
	C6	Coupling coefficient of environmental risk and management risk.

**Table 7 ijerph-20-04008-t007:** Three-factor coupling values for port hazardous chemical logistics risk.

Coupling Factor	Risk Coupling Value
$T_{3}^{1}$ (a,b,c)	0.0434
$T_{3}^{2}$ (a,b,d)	0.0916
$T_{3}^{3}$ (a,c,d)	0.0463
$T_{3}^{4}$ (b,c,d)	0.0319

**Table 8 ijerph-20-04008-t008:** Variables used in personnel–ship–management coupling system.

Variable Name		Actual Meaning of Variable
Horizontal variable	L1 (t)	Risk level indicator of personnel risk behavior sub-system.
	L2 (t)	Indicates the risk level index of the ship risk sub-system.
	L3 (t)	Indicates the risk level index of the environmental risk sub-system.
Constant	C1	Coupling coefficient between knowledge and quality of operators, illegal operation, and equipment failure.
	C2	Coupling coefficient between characteristics of hazardous chemicals, ship loading state, and establishment of safety system.
	C3	Coupling coefficient between equipment failure, aging wear of equipment and facilities, and illegal operation.
	C4	Coupling coefficient between operating knowledge and quality, personnel safety training, and safety education and education management.
	C5	Coupling coefficient between violation of regulations, safety culture, and establishment of education management and safety system.
	C6	Coupling coefficient between safety awareness, safety system establishment, and characteristics of hazardous chemicals.
	C7	Coupling coefficient between ship loading, monitoring and early warning investment, and safety education and education management.
	C8	Coupling coefficient between safety and education management, safety system establishment, and knowledge and quality of operators.
	C9	Indicates the initial value of risk level of illegal operation factors.
	C10	Indicates the initial value of risk level of operator’s knowledge and quality factors.
	C11	Indicates the initial value of risk level of safety awareness factor.
	C12	Initial value of risk level of equipment proficiency factor.
	C13	Indicates the initial value of risk level of characteristic factors of hazardous chemicals.
	C14	Indicates the initial value of risk level of ship load factor.
	C15	Indicates the initial value of risk level of aging and wear factors of equipment and facilities.
	C16	Indicates the initial value of the risk level of the equipment failure factor.
	C17	Indicates the initial value of risk level of personnel safety training factors.
	C18	Indicates the initial value of risk level of safety system establishment factor.
	C19	Indicates the initial value of risk level of monitoring and early warning input factors.
	C20	Indicates the initial value of risk level of safety culture and education management factors.

**Table 9 ijerph-20-04008-t009:** Coupling coefficient changes of the two schemes.

Scheme Name	Change Proportion	Post-Change Value
Current	0	0.149
Current	0	0.130
Current1	−30%	0.1043
Current1	−30%	0.091
Current2	+30%	0.1937
Current2	+30%	0.169

**Table 10 ijerph-20-04008-t010:** Changing the coupling coefficient under the two schemes.

Scheme Name	Change Proportion	Post-Change Value
Current	0	0.127
Current	0	0.160
Current1	+30%	0.1651
Current1	+30%	0.208
Current2	−30%	0.889
Current2	−30%	0.112

**Table 11 ijerph-20-04008-t011:** Coupling coefficient changes under the two schemes.

Scheme Name	Change Proportion	Post-Change Value
Current	0	0.127
Current	0	0.140
Current	0	0.151
Current1	−30%	0.0889
Current1	−30%	0.098
Current1	−30%	0.1057
Current2	+30%	0.1651
Current2	+30%	0.182
Current2	+30%	0.963

## Data Availability

We declare that all data, models, and code generated or used during the study appear in the submitted article.

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
