# Peer review of "Measurement and Simulation of Risk Coupling in Port Hazardous Chemical Logistics"

_ijerph, 2023, doi:10.3390/ijerph20054008_

Round 1

Reviewer 1 Report

The article concerns an interesting issue, which is the issue of risk coupling in port logistics of hazardous chemicals.

My comments and suggestions for development:

1. Introduction
- The Authors of the paper could consider clarifying the term "hazardous chemicals", which would make the paper easier to read and consider;
- In the study, the Authors often use the phrases "risk" (To what has the risk been referred?), "risk factors", these terms also require clarification;
- line 53 – no dot after "al.";
- lines 108, 110 – no space between name and parenthesis;
- line 155 – no space between 1) and Index
2. Analysis on the coupling effect (…)
- line 176 – no capital letter;
- Figure 1 - no information in the figure on which accidents the statistical data refer to (injured in accidents in general);
- lines 223, 230,239,243 - no spaces;
- lines 258-259 - it is worth referring to the source where the term was defined;
- line 353 - double dot.
3. Application of port hazardous (…)
- line 429 - table 4.4.
4. Simulation study on (...)
- Fig. 7-11 - double dot.
The study lacks a clear separation of the Material and Method chapters. The Authors also do not refer to legal regulations in this area.

Reviewer 2 Report

o   This paper aims a construction of the risk coupling system of port hazardous chemicals, logistics and analyze the coupling effect of the risk system. I highly recommend summarize figures and tables throughout the manuscript because are too extensive.  However, this paper has a major flaw because more than 80% contains statements without any references.

o   The references used in the manuscript are presented with names instead of surnames and not correctly cited after doble check the list of references. Some cases references are cited "et al" and "et al."

o   The aim in the introduction section is not clear and relevant information about statistic in China should be included to justify the study. This section is presented more like a discussion (line 46-130) and does not reflect the real problem which is chemical risks and their logistics. To my understanding it is not clear the meaning of this section in the context of paper. All this part needs a reorganization of the contents with a better contextualization and more logical connection among the section.

o   What it is AHP, (line 85)

o   Figure 1 does not bring any source, location, information

o   Figure 2 does not bring any information about this flow chart, reference

o   Figure 6 it’s not clear

o   Figure 11 it’s not clear

o   Figure 12, 13, 14 and 15 should be summarized

o   Which criteria the authors used to choose hazardous chemicals risk (line 2019).

o   The methodology used should be supported with updated references.

o   The authors are also highly requested to summarize the lack in literature that they find and that justify the development of this research.

o   The findings of the work may appear to be trivial if are not well discussed in the context of the extant body of literature. Authors can build a discussion to help readers to follow the logical development of the manuscript. I invite authors to build the arguments in favor of how this work contributes the construction of the risk coupling system of port hazardous chemicals, logistics and analyze the coupling effect of the risk system

o   The presented work has some merits, but at the same time, the provided future research directions and conclusions appear to be trivial.

Reviewer 3 Report

The submitted manuscript is resolving an important challenge connected with an accounting of risk impacts from hazardous chemicals' logistics into improving sustainability operation at the port

1) Please, add your affiliations;

2) The paper has some typos and grammar can be improved;

3) To better understand the importance and highlight the novelty of the studied problem, the literature review, in my opinion, can be wider. Because the presented list of references includes only research from Asian authors, this does not give a full saw of how this issue was previously resolved in other part of the world. Maybe better to add analyse results of European, African, and American according to considered tasks.

4) In line 370, I was confused because I find a citation [93], but the list of references contains only 36 publications. Also, this is very strange because I found the last citation before line 370 only in line 129 with quote [20]. To continue this issue, I note that I can’t found any other citations in brackets [] after line 370. Remember, the final references list has 36 unit. Please, fix this flaws;

5) The mathematical symbol “x” must be added in formulas (2-7) because in other cases (formulas) this operation is present;

6) In Figure 10, we can see an interesting tendency, that one graph (column (a, b, d)) from the histogram has a higher value than the other. Such tendency we found in figure 4 where one column (a, b) also be higher than others. In this case, the fact that column (a, b, c) in Figure 10 is smaller than column (a, b, d) looks like as strange. Because from comparing Figures 10 and 7, we can make conclusions that the main impact on obtained results have groups “a” and “b”. Therefore, the diagram columns where these two risk groups (a and b) are simultaneously compounded should be comparable. Perhaps a typo was made in the calculations, or this nuance should be additionally explained in the manuscript;

7) In my opinion, it will be better to change locations of the minus scale at ordinates axis (Y) on the opposite (Figure 9). Because according to a classic presentation of minus ordinates' axis (y), the scale will have the next gradation: -8,000; -2,000; -600; -400; 0 then put plus ordinate;

Finally, the paper, its novelty, structuration and obtained results impressed me.

Round 2

Reviewer 2 Report

Dear authors

The manuscript was improved according the revision and is suitable for publication.

Best Regards